# Stratified Prediction-Powered Inference for Hybrid Language Model Evaluation

**Adam Fisch**[†,∗]    **Joshua Maynez**[†,∗]    **R. Alex Hofer**[†]

**Bhuwan Dhingra**[†]    **Amir Globerson**[‡]    **William W. Cohen**[†]

[†]Google DeepMind    [‡]Google Research

`{fisch,joshuahm,rofer,bdhingra,amirg,wcohen}@google.com`

## Abstract

Prediction-powered inference (`PPI`) is a method that improves statistical estimates based on limited human-labeled data. `PPI` achieves this by combining small amounts of human-labeled data with larger amounts of data labeled by a reasonably accurate—but potentially biased—automatic system, in a way that results in tighter confidence intervals for certain parameters of interest (e.g., the mean performance of a language model). In this paper, we propose a method called Stratified Prediction-Powered Inference (`StratPPI`), in which we show that the basic `PPI` estimates can be considerably improved by employing simple data stratification strategies. Without making any assumptions on the underlying automatic labeling system or data distribution, we derive an algorithm for computing provably valid confidence intervals for population parameters (such as averages) that is based on stratified sampling. In particular, we show both theoretically and empirically that, with appropriate choices of stratification and sample allocation, our approach can provide substantially tighter confidence intervals than unstratified approaches. Specifically, `StratPPI` is expected to improve in cases where the performance of the autorater varies across different conditional distributions of the target data.

## 1 Introduction

Evaluating machine learning models requires *evaluation* data. In particular, to iteratively improve on a method during development, or to reliably report an improvement, one needs to confidently and quantitatively assess the performance of the method. This is especially challenging for large language models (LLMs), where gathering high-quality annotations for generations can be difficult and time-consuming—and can ultimately become quite costly if gathering more than a few hundred examples.

One often-proposed approach to avoiding the evaluation bottleneck is to use a secondary LLM-based system to judge the output of the primary one. For instance, if the primary task is developing an LLM-based question-answering (QA) system, one can use a second LLM-based system that rates question/answer pairs as acceptable or not [6; 8; 17]. However, automated raters (i.e., *autoraters*) may be biased relative to the human raters they are intended to model, as others have noted [1; 9; 18; 23]. This can become substantially worse when models are tailored to hill climb on the autorater metrics, and eventually cease to become a good metric at all—a phenomenon commonly referred to as Goodhart's law, or reward hacking in the context of reinforcement learning from human feedback [13; 24].

We thus have two signals for assessing model performance. The first is human labels, which are typically accurate, but expensive to collect. As a result, usually only a small sample size is available, and

---

∗Equal contribution.

an estimate based on these samples alone will have high variance. The second is autorater predictions, which are easy to collect for large sample sizes, but may also be systematically biased. The above may suggest that one must make a choice between either (i) a high-variance, but unbiased, estimate from a small human sample, or (ii) a lower-variance, but biased, autorater-based estimate. However, it turns out there are also statistically valid ways of *combining* the auto-rater and human data for hybrid evaluations. Following [1; 3; 7] we call such methods *prediction-powered inference* (PPI) methods. At a high level, PPI-based methods operate by using a small sample of examples labeled by both humans and autoraters to estimate the bias of the autorater. This bias is then used as a rectifier for the autorater estimate. The resulting estimate can then be shown to provide improved (i.e., tighter) confidence intervals for properties of interest for the system being evaluated, such as its true mean accuracy.

A weakness of standard PPI, however, is that it does not take heterogeneity into account. For example, in our QA setting, an autorater may have one accuracy when predicting if a model answer is correct, and a different accuracy when predicting if it is incorrect. This is especially true in cases where a correct answer is easy to verify (e.g., it is also present in Wikipedia), but harder to refute (e.g., no relevant external search results can be retrieved to either support or contradict it). In these settings, it may make sense to apply a different PPI strategy within each subdomain, depending on the local quality of the autorater. Moreover, we claim that such heterogenous settings are to be quite expected in practical applications. Inspired by the rich prior literature on stratified sampling and survey design [12; 19; 21], we therefore propose a *stratified* approach to PPI, and show that it can be very advantageous when performance varies across subdomains, for either the autorater, or the model being evaluated.

On a technical level, it is not immediately clear how to apply stratification to PPI, as it involves two types of samples: one sample that is labeled by both humans and an autorater, and another (typically much larger) sample that is only labeled by the autorater. Extending the analysis of [3], in this work we show how confidence intervals based on the asymptotic normality of weighted M-estimators [32] can in fact be derived for the stratified PPI setting. The next challenge we address is how to determine the sample sizes used for stratification (i.e., the sample size of each stratum). Similar to recent work for active statistical inference [37], we further derive optimal rates that depend on certain moments of the underlying distribution that are generally unknown—and provide an approach for effectively approximating these. Finally, we provide extensive empirical evidence showing that our stratified approach (StratPPI) leads to considerably improved confidence intervals over both classical inference methods that use only human labels, and the baseline (unstratified) PPI approach.

## 2 Related Work

Prediction Powered Inference (PPI) was introduced in [1] as a method for obtaining tighter confidence intervals for parameters learned in supervised machine learning (e.g., coefficients in logistic regression) by also leveraging other model-based predictions on additional, unlabeled data. Related ideas were explored by [30], but with a focus on bootstrapping as a way for obtaining confidence intervals. PPI was then extended in several directions. For example [36] showed how the labeled data can be used for both estimating the parameters and the autorater model. PPI++ [3] showed how to obtain confidence intervals that are easy to compute efficiently, and introduced a parameter for weighting the predictions of the autorater such that the overall statistical efficiency can be improved. As noted in previous works on PPI, these approaches are closely related to other statistical methods for introducing control variates based on autoevaluators [9], as well as augmented inverse propensity weighting [27] (see discussion in [1; 3]). Like this paper, prior work has also focused on using PPI/PPI++ for evaluating machine learning systems with autoraters, including for ranking chatbots in the Chatbot Arena [7] and evaluating retrieval-augmented generation systems [28].

Most relevant to our setting, the recent work of [37] focuses on *active* sampling of examples to label during PPI, where the total number of queried labels is random, but less than the sampling budget $n$ in expectation. Specifically, it proposes to label examples for which the autorater is less confident in its predictions, and corrects for the sampling bias with a variant of inverse propensity weighting. In contrast, our somewhat simplified approach takes inspiration from stratified sampling where a coarse stratification is defined in advance, and the total number of labeled samples is constant. Like [37], we derive an analogous optimal allocation strategy for a given budget, though we only apply it at the stratum level, and not for individual examples. Furthermore, while the variable allocation helps reduce variance in heterogenous settings, our stratified treatment also allows for a stratum-specific extension of [3]'s estimated tuning parameters that further improves performance in complementary ways.

In terms of stratification specifically, concurrent work [14] also proposed Bayesian variants of several `PPI` methods, including stratified approaches. In contrast, the methods here do not require introducing priors, do not require running expensive Bayesian inference, and give more conventional, frequentist, guarantees of performance. The credible intervals produced by Bayesian methods are related to, but different from, the confidence intervals produced here, and by other prior `PPI` approaches [1; 3; 7].

Finally, there is a conceptual similarity between the `PPI` rectifiers and post-hoc calibration of a classifier. There is a rich literature on calibrating classifiers (e.g., [22; 25]), but there is a clear difference in goals between calibration and `PPI`: the former is aimed at modifying a learned autorater to make better probabilistic predictions, and the latter is aimed at obtaining better confidence intervals by making use of an existing autorater. That said, clearly one approach to improving `PPI` is to use a better-calibrated classifier, perhaps by taking some of the labeled data and using it for re-calibration. This is similar in motivation to the cross-`PPI` approach of [36]. We leave such approaches as future work, but we do experimentally explore `PPI` approaches on both well-calibrated autoraters and poorly calibrated ones.

## 3 Preliminaries

We begin by briefly reviewing `PPI` [1], and specifically the efficient `PPI++` variant of [3]. Here, and in the rest of the paper, upper-case letters ($X$) denote random variables; lower-case letters ($x$) denote constants, and script letters ($\mathcal{X}$) denote sets, unless specified. All proofs are deferred to Appendix A.

Following [1; 3; 7], we assume an empirical sample $S_n = \{(X_1, Y_1), \ldots, (X_n, Y_n)\}$ of $n$ i.i.d. examples drawn from some unknown, but fixed, distribution $\mathbb{P}$, where $X_i \in \mathcal{X}$ is an input, and $Y_i \in \mathcal{Y}$ is the target output. We also assume a larger sample $\tilde{S}_N = \{(\tilde{X}_1, f(\tilde{X}_1)), \ldots, (\tilde{X}_N, f(\tilde{X}_N))\}$ where $N \gg n$, for which the target outputs are not available, but we have access to an *autorater function* $f \colon \mathcal{X} \to \mathcal{Y}'$ which provides an approximation of $Y$ given the observed input (we use $\mathcal{Y}'$ to denote the output space of the autorater as it is possible that $\mathcal{Y} \neq \mathcal{Y}'$, e.g., if $f(X) = \mathbb{E}[Y \mid X]$).

As an example, assume $X$ is a *(question, model answer)* tuple where the answer is produced by an LLM-based QA system we wish to evaluate, $Y$ is a $0/1$ "gold" human rating of correctness of the answer, i.e., $\mathbf{1}\{$*human rates model answer as correct*$\}$, and $f(X)$, the autorater, imperfectly predicts this human rating $Y$. One value of interest is then the average accuracy of the QA system, $\mathbb{E}[Y]$. More generally, given any convex loss function $\ell_\theta(x, y)$ satisfying certain regularity conditions (see Definition A.1), we are interested in estimating the $d$-dimensional parameter $\theta^*$,

$$\theta^* = \operatorname*{argmin}_{\theta \in \Theta} \mathbb{E}[\ell_\theta(X, Y)]. \tag{1}$$

In the statistics literature, this is broadly referred to as M-estimation (e.g., see [29]). Here, we focus on *mean* estimation, for which the loss $\ell_\theta(x, y) = \frac{1}{2}\|y - \theta\|^2$ has the optimum $\theta^* = \mathbb{E}[Y]$. However, our method generalizes to other typical losses such as the squared loss for linear regression, where the parameters $\theta^*$ are the regression coefficients, or, similarly, the log loss for other generalized linear models. Of course, the true $\theta^*$ is generally not known, because we cannot exactly calculate the expectation in (1). The goal of `PPI`, which we share in this work, is to use both the labeled and unlabeled data $S_n$ and $\tilde{S}_N$ to obtain an asymptotically valid confidence interval $\mathcal{C}_{\alpha,j}$ for $\theta_j^*$ that for any $j \in [d]$ satisfies[1]

$$\lim_{n, N \to \infty} \mathbb{P}(\theta_j^* \in \mathcal{C}_{\alpha,j}) \geq 1 - \alpha. \tag{2}$$

The simplest confidence interval can be obtained using standard techniques by using only the labeled sample $S_n$, and ignoring the autorater data $\tilde{S}_n$. However, `PPI` employs a clever trick which allows for also using the unlabeled sample $\tilde{S}_N$ to get tighter confidence intervals, as described next.

### 3.1 A rectified prediction-powered loss

The key idea of `PPI`-based methods is to use autorater predictions to derive a low variance, but unbiased, estimate of the objective in (1). Consider the following "rectified" prediction-powered loss:

$$L^{\mathrm{PP}}(\theta) = \underbrace{\frac{1}{N} \sum_{i=1}^{N} \ell_\theta(\tilde{X}_i, f(\tilde{X}_i))}_{\text{autorater data loss}} + \underbrace{\frac{1}{n} \sum_{i=1}^{n} \ell_\theta(X_i, Y_i) - \ell_\theta(X_i, f(X_i))}_{\text{loss rectifier}}. \tag{3}$$

---

[1]In $d$-dimensions, we obtain a confidence interval for each coordinate. See [3] for other possible choices.

The rectifier term on the right hand side of (3) removes the bias of the autorater data so that $L^{\mathrm{PP}}(\theta)$ satisfies $\mathbb{E}[L^{\mathrm{PP}}(\theta)] = \mathbb{E}[\ell_\theta(X, Y)]$. However, when $f(X)$ is correlated with $Y$, the loss $L^{\mathrm{PP}}(\theta)$ will have lower variance. Unfortunately, $f(X)$ may not be a good predictor of $Y$ in all cases. In fact, it is even possible for $L^{\mathrm{PP}}$ to be higher variance than the classical estimate (e.g., if $f(X)$ is anti-correlated with $Y$). Thus, to adapt to the quality of $f(X)$, PPI++ also introduces a tuning parameter $\lambda \in \mathbb{R}$:[2]

$$L_\lambda^{\mathrm{PP}}(\theta) = \frac{\lambda}{N} \sum_{i=1}^{N} \ell_\theta(\tilde{X}_i, f(\tilde{X}_i)) + \frac{1}{n} \sum_{i=1}^{n} \ell_\theta(X_i, Y_i) - \lambda \ell_\theta(X_i, f(X_i)). \tag{4}$$

Clearly, $L_\lambda^{\mathrm{PP}}$ still remains unbiased for any value of $\lambda$. For $\lambda = 0$, the framework reduces to classical inference. In most cases, however, a proper choice of $\lambda \neq 0$ will result in lower variance. PPI++ also suggests how to automatically choose a data-dependent $\hat{\lambda}$ that converges to an optimal value.

### 3.2 A prediction-powered confidence interval

We next describe the PPI++ method for deriving confidence intervals based on the rectified loss. Let

$$\hat{\theta}_{\hat{\lambda}}^{\mathrm{PP}} = \underset{\theta \in \Theta}{\arg\min} \, L_{\hat{\lambda}}^{\mathrm{PP}}(\theta). \tag{5}$$

be the prediction-powered estimate. Standard analysis for M-estimators can then be extended to show that the scaled difference of the estimate $\hat{\theta}_{\hat{\lambda}}^{\mathrm{PP}}$ and the true $\theta^*$ is asymptotically normally distributed. PPI++ then leverages this result to compute a confidence interval for $\theta^*$ that is asymptotically valid.

**Theorem 1** (PPI++, [3]). *Assume that $\hat{\lambda} \xrightarrow{p} \lambda$ and $\frac{n}{N} \to r \geq 0$. Let $H_{\theta^*} := \mathbb{E}[\nabla^2 \ell_{\theta^*}]$, and*

$$V_{f,\theta^*}^\lambda := \lambda^2 \mathrm{Cov}(\nabla \ell_{\theta^*}(\tilde{X}, f(\tilde{X})), \quad V_{\Delta,\theta^*}^\lambda := \mathrm{Cov}(\nabla \ell_{\theta^*}(X, Y) - \lambda \nabla \ell_{\theta^*}(X, f(X))), \tag{6}$$

*where $\lambda \in \mathbb{R}$ is a hyper-parameter. Then under the regularity conditions of Definition A.1, we have that $\sqrt{n}(\hat{\theta}_{\hat{\lambda}}^{\mathrm{PP}} - \theta^*) \xrightarrow{d} \mathcal{N}(0, \Sigma^\lambda)$, where $\Sigma^\lambda := H_{\theta^*}^{-1}(r \cdot V_{f,\theta^*}^\lambda + V_{\Delta,\theta^*}^\lambda)H_{\theta^*}^{-1}$.*

**Corollary 1** (PPI++ CI, [3]). *Let $\widehat{\Sigma}^{\hat{\lambda}}$ be the plug-in estimate for $\Sigma^\lambda$ using $S_n$ and $\tilde{S}_N$. Define*

$$\mathcal{C}_{\alpha,j}^{\mathrm{PP}} := \left( \hat{\theta}_{\hat{\lambda},j}^{\mathrm{PP}} \pm z_{1-\frac{\alpha}{2}} \sqrt{n^{-1}\widehat{\Sigma}_{jj}^\lambda} \right), \tag{7}$$

*where $z_\beta$ denotes the $\beta$-quantile of the standard normal distribution. Then for any $j \in [d]$ it holds that $\lim_{n,N \to \infty} \mathbb{P}\left(\theta_j^* \in \mathcal{C}_{\alpha,j}^{\mathrm{PP}}\right) \geq 1 - \alpha$.*

As mentioned above, [3] further showed that for an appropriate choice of $\lambda$ (that can be analytically derived and estimated via a $\hat{\lambda}$), the trace of the covariance matrix, $\mathrm{Tr}(\Sigma^\lambda)$, is at most that of the covariance matrix derived without using autorater data, implying that the PPI++-based confidence set $\mathcal{C}_\alpha^{\mathrm{PP}}$ can always be at least as tight as that of the classical M-estimator (and often much tighter in practice).

## 4 Stratified prediction-powered inference

We now present our approach for improving PPI++ estimates via *stratification*. In particular, we show how optimizing a rectified loss computed via stratified sampling can lead to a consistent, but lower variance, estimate of the optimal parameter $\theta^*$, and correspondingly tighter confidence intervals. Consider the QA example from the previous section. For most autoraters, it is reasonable to assume that the strength of their performance can vary, depending on the type of input being presented. For instance, an autorater might be accurate at predicting whether an answer to an unambiguous question (e.g., *"What is the capital of France?"*) is correct, but relatively poor at inferring if an answer to an open-ended question (e.g., *"What is the best way to cook a hamburger?"*) is acceptable or not. Splitting the problem space into different domains allows us to derive a more specialized form of the prediction powered loss that can better adapt to this autorater heterogeneity via stratified sampling.

Formally, assume that the input space $\mathcal{X}$ is partitioned in advance into $K$ non-empty, mutually exclusive, and exhaustive strata $\mathcal{A} = (\mathcal{A}_1, \ldots, \mathcal{A}_K)$, where $K$ is a finite integer. For each stratum,

---

[2]As noted in [3], for some losses $\ell_\theta$ we must have $\lambda \in [0, 1]$ to guarantee convexity of $L_\lambda^{\mathrm{PP}}$. The main focus of this paper, however, is on mean estimation with $\ell_\theta(x, y) = \frac{1}{2}\|y - \theta\|^2$, which is convex for any $\lambda \in \mathbb{R}$.

we further assume that we can estimate $w_k = \mathbb{P}(X \in \mathcal{A}_k)$ arbitrarily well using large amounts of unlabeled data or prior knowledge; we treat them as known constants here for simplicity. To collect the labeled and unlabeled datasets $S_n$ and $\tilde{S}_N$, we then follow a standard stratified sampling procedure in which we draw two i.i.d. sets of samples of fixed size $n_k$ and $N_k$, respectively, from each stratum $k$, where $\sum_{k=1}^{K} n_k = n$ and $\sum_{k=1}^{K} N_k = N$. The relative sizes $n_k/n$ and $N_k/N$ are free parameters; they can simply be the natural rates, $n_k/n \approx N_k/N \approx w_k$, or systematically decided (see §4.3). Note that this is a fundamentally different sampling model from standard `PPI`: here examples are i.i.d. within each strata, and independent (but not necessarily identically distributed) across each strata.

Next, we define the stratified loss via the weighted sum, $L_\lambda^{\mathrm{SPP}}(\theta) = \sum_{k=1}^{K} w_k L_{k,\lambda_k}^{\mathrm{PP}}(\theta)$, where $w_k$ is the stratum weight, $\lambda = (\lambda_1, \ldots, \lambda_k) \in \mathbb{R}^k$ are now *stratum-specific* tuning parameters, and $L_k^{\mathrm{PP}}(\theta)$ is the conditional prediction-powered loss computed within each stratum, i.e.,

$$L_{k,\lambda_k}^{\mathrm{PP}}(\theta) = \frac{\lambda_k}{N_k} \sum_{i=1}^{N_k} \ell_\theta(\tilde{X}_{ik}, f(\tilde{X}_{ik})) + \frac{1}{n_k} \sum_{i=1}^{n_k} \ell_\theta(X_{ik}, Y_{ik}) - \lambda_k \ell_\theta(X_{ik}, f(X_{ik})). \tag{8}$$

As before, each $L_{k,\lambda_k}^{\mathrm{PP}}$ is an unbiased estimate of the conditional loss. Like `PPI++`, we also allow for data-dependent parameters $\hat{\lambda}_k$ that we show how to automatically tune for best performance in §4.2.

### 4.1 A stratified prediction-powered confidence interval

We now present our main result, which is a confidence interval for $\theta^*$ based on the stratified loss. More precisely, the result states that, as in `PPI++`, the minimizer of the stratified loss,

$$\hat{\theta}_{\hat{\lambda}}^{\mathrm{SPP}} = \operatorname*{argmin}_{\theta} L_{\hat{\lambda}}^{\mathrm{SPP}}(\theta), \tag{9}$$

has an asymptotically normal distribution with mean $\theta^*$. See Algorithm 1 for pseudocode.

**Theorem 2.** *Assume that $\hat{\lambda}_k \xrightarrow{p} \lambda_k$, $\frac{n}{N} \to r$ for any $r \geq 0$, $\frac{n_k}{n} \to \rho_k$ for any $\rho_k > 0$, and $\frac{N_k}{N} \to \tilde{\rho}_k$ for any $\tilde{\rho}_k > 0$. Let $H_{k,\theta^*} := \mathbb{E}[\nabla^2 \ell_{\theta^*}(X, Y) \mid X \in \mathcal{A}_k]$, and*

$$V_{k,f,\theta^*}^{\lambda_k} := \lambda_k^2 \mathrm{Cov}(\nabla \ell_{\theta^*}(\tilde{X}, f(\tilde{X})) \mid \tilde{X} \in \mathcal{A}_k), \tag{10}$$

$$V_{k,\Delta,\theta^*}^{\lambda_k} := \mathrm{Cov}(\nabla \ell_{\theta^*}(X, Y) - \lambda_k \nabla \ell_{\theta^*}(X, f(X)) \mid X \in \mathcal{A}_k), \tag{11}$$

*where $\lambda_k \in \mathbb{R}$ is a hyper-parameter. Then, under the regularity conditions of Definition A.1,*

$$\sqrt{n}(\hat{\theta}_{\hat{\lambda}}^{\mathrm{SPP}} - \theta^*) \xrightarrow{d} \mathcal{N}(0, \Sigma_w^\lambda), \tag{12}$$

*where $\Sigma_w^\lambda := A_w^{-1} B_w^\lambda A_w^{-1}$ and:*

$$A_w := \sum_{k=1}^{K} w_k H_{k,\theta^*} \qquad B_w^\lambda := \sum_{k=1}^{K} w_k^2 \left( \frac{r}{\tilde{\rho}_k} \cdot V_{k,f,\theta^*}^{\lambda_k} + \frac{1}{\rho_k} \cdot V_{k,\Delta,\theta^*}^{\lambda_k} \right). \tag{13}$$

To obtain the result, we combine unstratified `PPI++` with the asymptotic properties of weighted M-estimators [32]. The resulting confidence interval for $\theta^*$ is then derived analogously to Corollary 1.

**Corollary 2.** *Let $\widehat{\Sigma}_w^{\hat{\lambda}}$ be the plug-in estimate for $\Sigma_w^\lambda$ using $S_n$ and $\tilde{S}_N$. Define*

$$\mathcal{C}_{\alpha,j}^{\mathrm{SPP}} := \left( \hat{\theta}_{\hat{\lambda},j}^{\mathrm{SPP}} \pm z_{1-\frac{\alpha}{2}} \sqrt{n^{-1} \widehat{\Sigma}_{w,jj}^{\hat{\lambda}}} \right), \tag{14}$$

*where $z_\beta$ denotes the $\beta$-quantile of the standard normal distribution. Then for any $j \in [d]$,*

$$\lim_{n,N \to \infty} \mathbb{P}\left(\theta_j^* \in \mathcal{C}_{\alpha,j}^{\mathrm{SPP}}\right) \geq 1 - \alpha. \tag{15}$$

The form of the stratified prediction-powered confidence interval is similar to that of `PPI++`, except that it is based off of the weighted stratum-conditional covariance matrices. The effect of this change, however, is significant. In fact, in the case of mean estimation, we show that the asymptotic variance of `StratPPI` is at most that of `PPI++` (even without any additional tuning of $\lambda_k$ and $\rho_k$).

---

**Algorithm 1** Stratified prediction-powered inference for general M-estimators (`StratPPI`)

---

**Definitions:** $f$ is the autorater. Inputs $\{(\tilde{X}_{ik}, f(\tilde{X}_{ik}))\}_{i=1}^{N_k}$ include the autorater predictions on sampled unlabeled data for each partition $\mathcal{A}_k$, $k \in [K]$. Inputs $\{(X_{ik}, Y_{ik}, f(X_{ik}))\}_{i=1}^{n_k}$ include the autorater predictions on sampled labeled data for each partition $\mathcal{A}_k$, $k \in [K]$. $w_k$ is the partition weight for $\mathcal{A}_k$, $k \in [K]$. $\alpha$ is the confidence interval coverage error tolerance.

1: # Pick weighting parameters, see §4.2.
2: Select $\hat{\lambda} = (\hat{\lambda}_1, \ldots, \hat{\lambda}_K)$
3: # Solve for the minimizer of the stratified, prediction-powered empirical loss.
4: $\hat{\theta}_{\hat{\lambda}}^{\mathrm{SPP}} = \operatorname{argmin}_\theta L_{\hat{\lambda}}^{\mathrm{SPP}}(\theta)$
5: # Estimate the Hessian of the true expected loss at $\theta^*$.
6: $\hat{A}_w = \sum_{k=1}^{K} \frac{w_k}{n_k} \sum_{i=1}^{n_k} \nabla^2 \ell_{\hat{\theta}_{\hat{\lambda}}^{\mathrm{SPP}}}(X_{ik}, Y_{ik})$
7: # Build the estimated covariance matrix from each stratified component of the loss.
8: $\widehat{\Sigma}_w^{\hat{\lambda}} = \mathbf{0}_{d \times d}$
9: **for** $k = 1, 2, \ldots, K$ **do**
10: $\qquad \widehat{V}_f = \hat{\lambda}_k^2 \widehat{\mathrm{Cov}}_{N_k} \left( \nabla \ell_{\hat{\theta}_{\hat{\lambda}}^{\mathrm{SPP}}}(\tilde{X}_{ik}, f(\tilde{X}_{ik})) \right)$
11: $\qquad \widehat{V}_\Delta = \widehat{\mathrm{Cov}}_{n_k} \left( \nabla \ell_{\hat{\theta}_{\hat{\lambda}}^{\mathrm{SPP}}}(X_{ik}, Y_{ik}) + \hat{\lambda}_k \nabla \ell_{\hat{\theta}_{\hat{\lambda}}^{\mathrm{SPP}}}(X_{ik}, f(X_{ik})) \right)$
12: $\qquad \widehat{\Sigma}_w^{\hat{\lambda}} = \widehat{\Sigma}_w^{\hat{\lambda}} + w_k^2 \hat{A}_w^{-1} \left( \frac{\widehat{V}_f}{N_k} + \frac{\widehat{V}_\Delta}{n_k} \right) \hat{A}_w^{-1}$
13: # Return coordinate-wise confidence intervals for $\theta^*$.
14: $\mathcal{C}_\alpha^{\mathrm{SPP}} = \left\{ \hat{\theta}_j^{\mathrm{SPP}} \pm z_{1-\frac{\alpha}{2}} \sqrt{\widehat{\Sigma}_{w,jj}^{\hat{\lambda}}} : j \in [d] \right\}$

---

**Proposition 1.** *Let $\lambda_k \in \mathbb{R}$ be any constant for all strata, and fix $\rho_k$ and $\tilde{\rho}_k$ to their natural rates $w_k$. Then for $\ell_\theta(x, y) = \frac{1}{2}\|y - \theta\|^2$ and any stratification $(\mathcal{A}_1, \ldots, \mathcal{A}_K)$, we have $\mathrm{Tr}(\Sigma_w^\lambda) \leq \mathrm{Tr}(\Sigma^\lambda)$, where $\Sigma^\lambda$ is the asymptotic covariance matrix of `PPI++`. Furthermore, we have that equality holds if and only if both $\mathbb{E}[Y \mid X \in \mathcal{A}_k]$ and $\mathbb{E}[f(X) \mid X \in \mathcal{A}_k]$ are the same for all strata.*

More generally, although our results will hold for arbitrary stratifications, it is best if they are heterogeneous, and chosen such that the individual stratum-conditional variances are minimized. For example, if an autorater systematically *over-estimates* the model's performance on one subdomain, but systematically *under-estimates* the model's performance on another, then splitting these subdomains into different strata can result in a lower variance $L_\lambda^{\mathrm{SPP}}$. Similarly, if an autorater has much higher noise on some subdomains than others, it can be beneficial to stratify on those subdomains—and then either lower $\lambda_k$, allocate a higher proportion of samples $n_k/n$ and $N_k/N$, or both for the noisier stratas. In §5 we empirically demonstrate that the stratified estimator can indeed lead to considerably tighter confidence intervals in practice, especially with additional tuning of $\lambda_k$ and $\rho_k$, as discussed next.

### 4.2 Optimal weighting of the autorater predictions

In [3] it was shown that the optimal value of $\lambda$ for `PPI++` (i.e., the one minimizing the variances of the estimator and the corresponding confidence interval) could be found in closed form. We now present a simple extension of this result to the stratified case. For notational convenience, we use the shorthand $\nabla \ell_{k,\theta} := \nabla \ell_\theta(X, Y) \mid X \in \mathcal{A}_k$ and $\nabla \ell_{k,\theta}^f := \nabla \ell_\theta(X, f(X)) \mid X \in \mathcal{A}_k$.

**Proposition 2.** *Assume that $\tilde{\rho}_k$ and $\rho_k$ are fixed. Then the tuning parameters $(\lambda_1^*, \ldots, \lambda_k^*)$, where*

$$\lambda_k^* = \frac{\mathrm{Tr}\left(A_w^{-1}(\mathrm{Cov}(\nabla \ell_{k,\theta^*}, \nabla \ell_{k,\theta^*}^f) + \mathrm{Cov}(\nabla \ell_{k,\theta^*}^f, \nabla \ell_{k,\theta^*}))A_w^{-1}\right)}{2\left(1 + \frac{n_k}{N_k}\right)\mathrm{Tr}\left(A_w^{-1}\mathrm{Cov}(\nabla \ell_{k,\theta^*}^f)A_w^{-1}\right)}, \tag{16}$$

*minimize the cumulative asymptotic variance, $\mathrm{Tr}(\Sigma_w^\lambda)$.*

Note that $\mathrm{Tr}(\Sigma_w^\lambda)$ is proportional to the total size of the confidence interval $\mathcal{C}_\alpha^{\mathrm{SPP}}$ in (14). Furthermore, as in [3], we can use plug-in estimates for the terms in (16) to compute a $\hat{\lambda}_k$, where $\hat{\lambda}_k \xrightarrow{p} \lambda_k^*$. From (16), we can see that $\lambda_k^*$ is closely related to the correlation coefficient of the (curvature-scaled)

gradients for minimizing the loss on an autorater label versus a true label. Intuitively, the more correlated these terms are, the more we can rely on the autorater labels for finding the true minimizer of $\mathbb{E}[\ell_\theta(X, Y)]$. For 1-$d$ mean estimation in particular, we can see that $\lambda_k^*$ takes on a simple form:

**Example 1** ($\lambda_k^*$ for mean estimation). *Consider the* 1-$d$ *mean loss:* $\ell_\theta(x, y) = \frac{1}{2}(y - \theta)^2$. *Then:*

$$\lambda_k^* = \frac{\mathrm{Cov}(Y, f(X))}{(1 + \frac{n_k}{N_k})\mathrm{Var}(f(X))} \approx \frac{\mathrm{Cov}(Y, f(X))}{\mathrm{Var}(f(X))} \quad \text{for large } N_k, \tag{17}$$

*which is equivalent to the optimal linear regression coefficient,* $\min_{\lambda_k} \mathbb{E}[|Y - \lambda_k f(X)|^2 \mid X \in \mathcal{A}_k]$.

### 4.3 Optimal allocation of the sampling budget

Our stratification approach has an additional hyperparameter $\rho$ that can also be tuned to reduce variance. Recall that $\rho_k$ determines the ratio between the labeled data size $n_k$ for stratum $k$ and the overall data size $n$ (i.e., $\sum_k n_k = n$). $\tilde{\rho}_k$ is similarly defined for the unlabeled data. Any strictly positive values of $\rho_k$ and $\tilde{\rho}_k$ are valid to be used, though not all values will improve performance. It turns out that the optimal $\rho_k$ values can be exactly calculated, as the following proposition shows.

**Proposition 3.** *Assume that* $\lambda_k$ *is fixed. Then the sampling rates* $(\rho_1^*, \ldots, \rho_k^*)$ *and* $(\tilde{\rho}_1^*, \ldots, \tilde{\rho}_k^*)$, *where*

$$\rho_k^* = \frac{w_k \sqrt{\mathrm{Tr}(A_w^{-1} V_{k,\Delta,\theta^*}^{\lambda_k} A_w^{-1})}}{\sum_{k'=1}^K w_{k'} \sqrt{\mathrm{Tr}(A_w^{-1} V_{k',\Delta,\theta^*}^{\lambda_{k'}} A_w^{-1})}} \quad \text{and} \quad \tilde{\rho}_k^* = \frac{w_k \sqrt{\mathrm{Tr}(A_w^{-1} V_{k,f,\theta^*}^{\lambda_k} A_w^{-1})}}{\sum_{k'=1}^K w_{k'} \sqrt{\mathrm{Tr}(A_w^{-1} V_{k',f,\theta^*}^{\lambda_{k'}} A_w^{-1})}} \tag{18}$$

*minimize the cumulative asymptotic variance,* $\mathrm{Tr}(\Sigma_w^\lambda)$.

Although the solution for $\rho_k^*$ is informative, it is not necessarily practical, as it depends on knowing $A_w^{-1} V_{k,\Delta,\theta^*}^{\lambda_k} A_w^{-1}$; this in turn depends on $\theta^*$ and $\mathbb{P}(X, Y)$, which are both unknown.[3] In the special case of mean estimation, however, it turns out there is no dependence on $\theta^*$. To address the remaining dependence on $\mathbb{P}(X, Y)$, we propose to use autorater confidence scores, assuming they are available. Specifically, assume $\mathcal{Y}$ is discrete, and let $c(y \mid x)$ be the confidence of the autorater in label $y$ given input $x$, where $c(y \mid x)$ approximates $\mathbb{P}(Y = y \mid X = x)$. This will result in the estimate for $\mathrm{Tr}(A_w^{-1} V_{k,\Delta,\theta^*}^{\lambda_k} A_w^{-1})$ below, which can then be plugged into the expression for $\rho_k^*$ in Proposition 3.

**Example 2** ($\rho_k^*$ for mean estimation). *Consider the* 1-$d$ *mean loss:* $\ell_\theta(x, y) = \frac{1}{2}(y - \theta)^2$. *Then:*

$$\mathrm{Tr}(A_w^{-1} V_{k,\Delta,\theta^*}^{\lambda_k} A_w^{-1}) = \mathrm{Var}(Y - \lambda_k f(X) \mid X \in \mathcal{A}_k). \tag{19}$$

(19) *can then be estimated using the observed, but unlabeled, samples scored by the autorater for each stratum* $k$, $\tilde{X}_{ik} = \tilde{x}_{ik}$, $i = 1, \ldots, N_k$, *as* $\mathrm{Var}(Y - \lambda_k f(X) \mid X \in \mathcal{A}_k) \approx \hat{\sigma}_k^2$, *where*

$$\hat{\sigma}_k^2 = \frac{1}{N_k} \sum_{i=1}^{N_k} \sum_{y \in \mathcal{Y}} c(y \mid \tilde{x}_{ik}) (y - \lambda_k f(\tilde{x}_{ik}) - \hat{\mu}_k)^2 \quad \text{and} \tag{20}$$

$$\hat{\mu}_k = \frac{1}{N_k} \sum_{i=1}^{N_k} \sum_{y \in \mathcal{Y}} c(y \mid \tilde{x}_{ik}) (y - \lambda_k f(\tilde{x}_{ik})). \tag{21}$$

For $d$-dimensional data, the result is similar, but with a sum of $d$ variances (one for each dimension). We also provide a simplified expression for $\hat{\sigma}_k^2$ in Appendix B when $f(\tilde{x}) := \sum_{y \in \mathcal{Y}} c(y \mid \tilde{x}) \cdot y$. Importantly, as we are free to use any $\rho_k > 0$, using this estimate still preserves the asymptotic coverage guarantees in (14), regardless of if the confidence estimate $c(y \mid x)$ is calibrated or not. Empirically, we show that using this heuristic can indeed lead to substantial improvements.

## 5 Experimental results

We compare our stratified estimator, `StratPPI`, to two baselines: (i) the classical estimate, which uses only the labeled data, $S_n$; and (ii) `PPI++`, which uses both $S_n$ and $\tilde{S}_n$. All of our experiments focus on

---

[3]Similarly, the optimal solution to $\tilde{\rho}_k$ also depends on the unknown $\theta^*$ in general, though this term is less important to optimize if we assume $N$ to be large. In practice, we always keep $\tilde{\rho}_k$ fixed to the natural rate, $w_k$.

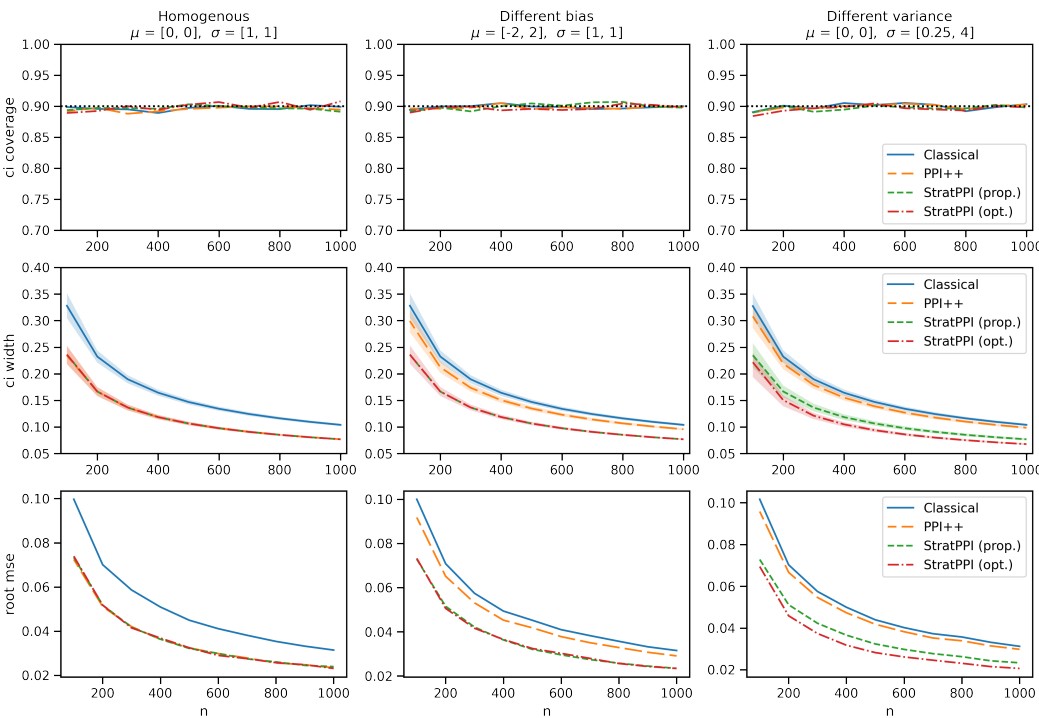

Figure 1: Mean estimation simulation study with $K = 2$ and $\alpha = 0.1$. The top row plots coverage (i.e., the fraction of the cases where the CI contained the true parameter value $\theta^*$). The middle row plots the mean CI width ($\downarrow$ is better). Shaded areas plot the $16/84$ quantiles across 5k trials. The bottom row plots the RMSE of $\hat{\theta}^{\mathrm{SPP}}$ computed across the $5k$ trials, which shares the same trend with the mean CI width, as the estimator is unbiased. The left column shows a setting where strata are homogeneous, and `StratPPI` provides the no benefits over standard `PPI++` (but is not worse). The middle and right columns show heterogeneous settings where the autorater has either a different bias ($\mu$) or variance ($\sigma$) per stratum, in which case `StratPPI` helps substantially. As strata variances are known, we only report proportional and optimal sample allocation results for `StratPPI`.

1-$d$ mean estimation. We explore three different allocation strategies for `StratPPI`: the first is to set $\rho_k = w_k$ to be data proportional (`StratPPI Prop.`), the second is to set $\rho_k$ optimally via the oracle $\rho_k = \rho_k^*$ (`StratPPI Opt.`), and the third is to use the approximation, $\rho_k \propto w_k \hat{\sigma}_k$, in Example 2 for $\lambda_k = 1$ when confidence scores are available (`StratPPI Heur.`). We use $\lambda$-tuning for both `PPI++` and `StratPPI`, as outlined in §4.2. Additional experimental results are given in Appendix C.

## 5.1 Simulation studies

We start with a simple synthetic experiment that is an analogue of §7.7.1 in [3]. Our goal is to estimate the mean outcome $\mathbb{E}[Y]$, where $Y \sim \mathcal{N}(0, 1)$. We assume that the input space $\mathcal{X}$ is partitioned into $K = 2$ strata, $(\mathcal{A}_1, \mathcal{A}_2)$, of equal mass $\mathbb{P}(X \in \mathcal{A}_1) = \mathbb{P}(X \in \mathcal{A}_2) = 0.5$. We then assume that predictions are formed as $f(X_{ik}) = Y_{ik} + \mu_k + \sigma_k \epsilon_{ik}$, where $\epsilon_{ik} \sim \mathcal{N}(0, 1)$. In other words, the predictions do not depend on the covariates $X_{ik}$, other than to reflect a stratum-specific noise $\sigma_k$ and bias $\mu_k$. We test three different scenarios: (i) where the two strata are homogeneous with $\mu_1 = \mu_2$ and $\sigma_1 = \sigma_2$; (ii) where the two strata have different prediction biases, $\mu_1 \neq \mu_2$; and (iii) where the two strata have different prediction noise levels, $\sigma_1 \neq \sigma_2$. For each experiment, we sample $N = 10{,}000$ total predictions $f(\tilde{X})$ using $\tilde{\rho}_1 = \tilde{\rho}_2 = 0.5$, i.e., proportional to masses of the two hypothetical, equal-weight strata. We then vary the total number $n$ of labeled examples $Y$, where the allocation is chosen according to $\rho$ (which differs depending on if we are using `StratPPI Prop.` or `StratPPI Opt.`). We show results in Figure 1 for the mean confidence interval (CI) size and coverage (i.e., the fraction of the cases where the CI contained the true parameter value $\theta^*$) of each method, averaged over $1k$ trials. As the plots in Figure 1 illustrate, when the underlying strata are homogeneous (left column), `StratPPI` behaves similar to `PPI++`. However, when the strata are heterogeneous (middle and right columns), `StratPPI` outperforms both baselines significantly—whereas `PPI++` becomes barely

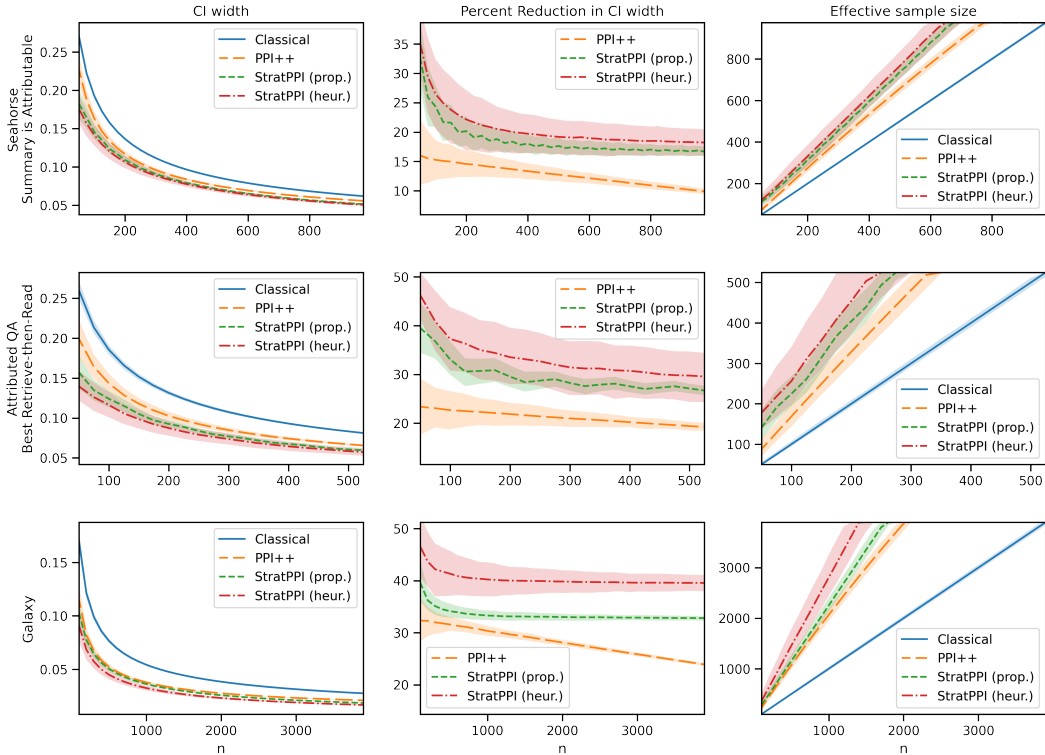

Figure 2: Mean estimation on real data with $K = 10$ and $\alpha = 0.05$. The $x$ axis plots the number of human-labeled examples $n$; the $y$ axis plots CI width, percent reduction in CI width against the classical estimate , and the effective sample size (the amount of human labels necessary to match the same confidence interval via classical inference). Shaded areas plot the $16/84$ quantiles across $1k$ trials. All `StratPPI` methods improve over classical inference and `PPI++`.

more powerful than the classical estimator. Additionally, we see that when the variance differs per stratum (right column), optimal allocation of the sampling budget indeed provides additional benefit.

## 5.2 Real data studies

We now demonstrate how our method performs on real datasets, where the underlying structure of the autorater is unknown. To partition $\mathcal{X}$, we choose to focus on stratifications that are based on the autorater's predictions, $f(X)$, based on the intuition that autorater performance can often differ across the type of predictions that it makes. Concretely, if the output space $\mathcal{Y}'$ of $f$ is discrete, then we define $\mathcal{A} = \mathcal{Y}'$; otherwise we define $\mathcal{A}$ based on the equal-mass quantiles of $\mathcal{Y}'$ (which can be estimated by sampling a large set of unlabeled $X$ and applying the autorater). We set $K = 10$. For all experiments, we plot performance as a function of $n$, where $n$ is the number of human ratings our system is allowed to observe from each dataset. The remainder of the dataset (including any data points that are unlabeled, or labeled but with the labels removed) is used for the autorater sample $\tilde{S}_N$.

**Seahorse.** The `Seahorse` dataset [11] focuses on multilingual summarization. The authors considered generative models that output summaries for a document, and collected labels for many systems that cover serveral dimensions of summary quality. We focus on one quality dimension—whether the summary is fully attributable to the source document—and on one summarization system—a fine-tuned 13B parameter mT5 model [33]. The autorater models for each dimension are also mT5-XXL finetuned models, which output probability scores. The data contains 2727 examples for these two tasks, all of which have both human ratings as well as autorater scores.

**AttributedQA.** In attributed question answering [6], the goal of the QA system is to output both an answer, and a retrieved document that provides support for that answer. The system is only considered to be correct if the answer is both correct, and indeed supported by the linked document. We evaluate the highest-scoring "retrieve-and-read" QA system from this dataset,and define our autorater to be an 11B parameter T5 model [26] fine-tuned on a collection of natural language

entailment tasks [15]. Like `Seahorse`, the model predicts a probability for whether or not the QA system gave an attributable answer. This dataset has 1000 human labels and 3000 autorater labels.

**Galaxy.** To demonstrate the generality of `StratPPI` beyond LLM-based settings, we also consider the `Galaxy` dataset [31], where the task is to estimate the fraction of spiral galaxies in the local universe. The autorater used here is a ResNet classifier applied to images from the Sloan Digital Sky Survey (SDSS) [34]. The classifier estimates the probability that the galaxy in question is spiral. We use 16,743 observations from the dataset which contain both the human and autorater labels.

**Methodology.** We follow a procedure similar to [1; 3] to study CI estimates as a function of $n$. We report results on the following estimates: Classical inference, PPI++, `StratPPI Prop.` and `StratPPI Heur`. We do not report `StratPPI Opt.` since it is unknown for real data. For each value of $n$, we sample $n$ of the cases with human labels at an allocation rate $\rho_k$ (this rate is determined differently in `StratPPI Prop.` and `StratPPI Heur.`). As noted above, we construct the unlabeled dataset $\tilde{S}_N$ by joining the remaining labeled data (without utilizing the true labels) with any unlabeled data available for that dataset. We repeat this over 1000 trials, and for each trial we obtain the CI width. We report the mean of these widths in Figure 2, as well as the percent reduction in width over the classical inference baseline, $\frac{|\mathcal{C}_\alpha^{\text{classical}}| - |\mathcal{C}_\alpha^{\text{method}}|}{|\mathcal{C}_\alpha^{\text{classical}}|} \times 100\%$. We also report the "effective sample size", which we define as the number of samples required to obtain a CI of the same width as the method at sample size $n$ when using the classical inference baseline instead.

**Results.** Figure 2 shows a large improvement for `StratPPI` methods over both PPI++ and classical inference for most datasets. Specifically, though all CIs decrease substantially in absolute size with $n$ as expected, we see that improvements in the percent reduction in CI width over PPI++ can be as large as $0.10 \rightarrow 0.20$, $0.20 \rightarrow 0.30$, and $0.25 \rightarrow 0.35$ points in `Seahorse`, `AttributedQA`, and `Galaxy`, respectively. Furthermore, we can observe that many of the datasets exhibit heterogeneous characteristics, for which heuristic allocation helps considerably. In `Galaxy`, this accounts for a $+10$ percent reduction in CI width. In `Seahorse` and `AttributedQA`, the improvements are less strong but still clearly apparent. The practical implication of these results is that when limited by a human rating budget, `StratPPI` is able to produce an estimate of the mean with fewer human ratings via stratification and sampling allocation. For example, for the `Seahorse` dataset, we can see from the right column in Figure 2 that `StratPPI Heur.` with only 300 human ratings will be approximately as confident about the mean as the classical estimate that utilizes 600 human ratings, a factor of $2\times$.

## 6    Conclusion

As systems built on top of large language models continue to become more and more advanced, it becomes increasingly challenging to evaluate their performance using automatic tools. Manual labeling, on the other hand, is slow and expensive. Methods which therefore save on annotation cost are critical for reliably evaluating models, and knowing when they are improving—or degrading. Prediction-powered inference (PPI) is a promising class of such hybrid evaluation methods, since it can leveraged to provably produce statistically valid confidence intervals, while also effectively reducing the number of human labels needed to obtain intervals of certain width. Our results demonstrate that we can push PPI even further by introducing a method for performing even lower variance M-estimation by employing stratified sampling. In particular, we find that stratifications based on the predictions of the autoraters themselves proves to be an powerful stratification technique.

**Limitations.** While the confidence intervals produced by `StratPPI` (and PPI) enjoy coverage guarantees, these guarantees are asymptotic. When finite-sample performance is of particular importance, techniques that afford stronger guarantees might be preferred [2; 4; 5]. We also note some non-trivial aspects of the stratified sampling setup: (i) the number of strata has to be fixed; if $K$ scales with $n$ a more careful treatment is required; and (ii) the assumed observation model is different from traditional i.i.d. settings—we must be able to sample fixed sized samples from each partition; and (iii) performance may not be improved if the selected stratification is not statistically useful. Furthermore, in practice, the human data might have already been collected, in which case the studied stratified sampling setup does not directly apply. We leave the study of such post-stratified estimators to future work.

**Broader impacts.** This paper introduces a more powerful statistical method for evaluating LLMs, by merging human evaluations with autoraters in a way that is aware of subdomain differences. Our goal is to help power more reliable evaluations with lower annotation effort, both in terms of cost and time.

**Acknowledgements.** We thank Jacob Eisenstein, Jonathan Berant, Anastasios Angelopoulos, Taylan Cemgil, Arnoud Doucet, and Chris Dyer for helpful comments and discussions.

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

# A  Proofs

We begin by defining the basic regularity conditions that we assume $\ell_\theta$ satisfies:

**Definition A.1** (Regularity conditions of $\ell_\theta$). *Assume that*

  (i) $\Theta$ *is a compact subset of* $\mathbb{R}^d$;

  (ii) *The minimizer* $\theta^* \in \text{int}(\Theta)$ *is unique with* $\mathbb{E}[\nabla \ell_{\theta^*}(X, Y)] = 0$;

  (iii) *For all* $(x, y) \in \mathcal{X} \times \mathcal{Y}$, $\ell_\theta(x, y)$ *is twice continuously differentiable on* $\text{int}(\Theta)$;

  (iv) *For all* $\theta \in \Theta$, $\ell_\theta(X, Y)$, $\frac{\partial \ell_\theta(X,Y)}{\partial \theta_i}$, *and* $\frac{\partial^2 \ell_\theta(X,Y)}{\partial \theta_i \partial \theta_j}$ *have finite expectations;*

  (v) $\mathbb{E}[\nabla^2 \ell_\theta(X, Y)]$ *is non-singular.*

These regularity conditions are fairly mild—for example, it is straightforward to verify that the loss of the mean estimator, $\frac{1}{2}\|y - \theta\|^2$, satisfies Definition A.1. We note, however, that `PPI++` can be applied to merely "smooth enough" losses, which also include non-continuously differentiable functions losses like the quantile loss. As we primarily focus on mean estimation in this work, for simplicity of analysis we only consider (the still broad class) of losses satisfying Definition A.1, though our results can be expected to readily extend to the more general case following a similar treatment as in [3; 29].

To support our theoretical results, we also provide the following two lemmas.

**Lemma A.1** (Slutsky's Theorem, general form). *Let* $X_n \xrightarrow{d} X$ *and* $Y_n \xrightarrow{p} c$, *where* $X_n$, $X$, *and* $Y_n$ *are random vectors, and* $c$ *is a constant vector. Then for any continuous function* $g$, $g(X_n, Y_n) \xrightarrow{d} g(X, c)$.

*Proof.* By Theorem 2.7 of [29] we have $(X_n, Y_n) \xrightarrow{d} (X, c)$. The continuous mapping theorem then implies that $g(X_n, Y_n) \xrightarrow{d} g(X, c)$. $\qquad\square$

**Lemma A.2.** *Under the regularity conditions of Definition A.1, we have that* $\hat{\theta}_{\hat{\lambda}}^{\text{SPP}} \xrightarrow{p} \theta^*$.

*Proof.* Under the regularity conditions of Definition A.1 and the fact that $\hat{\lambda}_k \xrightarrow{p} \lambda_k$ where $\lambda_k$ is constant, the uniform weak law of large numbers can be applied to each term in $L_{\hat{\lambda}}^{\text{SPP}}(\theta)$ so that

$$\sup_{\theta \in \Theta} \left\| L_{\hat{\lambda}}^{\text{SPP}}(\theta) - \mathbb{E}[\ell_\theta(X, Y)] \right\| \xrightarrow{p} 0.$$

As $\theta^*$ is unique, $\Theta$ is compact, and $\ell_\theta$ is continuous it follows from Theorem 2.1 of [20] that

$$\hat{\theta}_{\hat{\lambda}}^{\text{SPP}} \xrightarrow{p} \theta^*.$$

$\qquad\square$

## A.1  Proof of Theorem 2

*Proof.* For ease of notation, we will define

$$\tilde{L}_{N_k}^f(\theta) = \frac{1}{N_k} \sum_{i=1}^{N_k} \ell_\theta(\tilde{X}_{ik}, f(\tilde{X}_{ik})$$

$$L_{n_k}(\theta) = \frac{1}{n_k} \sum_{i=1}^{n_k} \ell_\theta(X_{ik}, Y_{ik})$$

$$L_{n_k}^f(\theta) = \frac{1}{n_k} \sum_{i=1}^{n_k} \ell_\theta(X_{ik}, f(X_{ik})).$$

We will also use the following shorthand for the conditional gradients:

$$\nabla \ell_{k,\theta} := \nabla \ell_\theta(X, Y) \mid X \in \mathcal{A}_k \quad \text{and}$$

$$\nabla \ell_{k,\theta}^f := \nabla \ell_\theta(X, f(X)) \mid X \in \mathcal{A}_k.$$

As samples are i.i.d. within each stratum, the CLT gives

$$\sqrt{N_k}\left(\nabla \tilde{L}_{N_k}^f(\theta^*) - \mathbb{E}[\nabla \ell_{k,\theta^*}^f]\right) \xrightarrow{d} \mathcal{N}\left(0, \mathrm{Cov}\left(\nabla \ell_{k,\theta^*}^f\right)\right), \tag{22}$$

and

$$\sqrt{n_k}\begin{bmatrix} \nabla L_{n_k}(\theta^*) - \mathbb{E}[\nabla \ell_{k,\theta^*}] \\ \nabla L_{n_k}^f(\theta^*) - \mathbb{E}[\nabla \ell_{k,\theta^*}^f] \end{bmatrix} \xrightarrow{d} \mathcal{N}\left(\begin{bmatrix} 0 \\ 0 \end{bmatrix}, \mathrm{Cov}\left(\begin{bmatrix} \nabla \ell_{k,\theta^*} \\ \nabla \ell_{k,\theta^*}^f \end{bmatrix}\right)\right). \tag{23}$$

Since samples $S_n$ and $\tilde{S}_N$ are independent from each other, we also have that (22) and (23) converge jointly. Applying Lemma A.1 for the following continuous function of $\hat{\lambda}_k \xrightarrow{p} \lambda_k$ then gives

$$\sqrt{n}\left(\hat{\lambda}_k \cdot \mathcal{N}\left(\nabla \tilde{L}_{N_k}^f(\theta^*) - \mathbb{E}[\nabla \ell_{k,\theta^*}^f]\right) + \begin{bmatrix} 1 \\ -\hat{\lambda}_k \end{bmatrix}^\top \begin{bmatrix} \nabla \tilde{L}_n(\theta^*) - \mathbb{E}[\nabla \ell_{k,\theta^*}] \\ \nabla \tilde{L}_{n_k}^f(\theta^*) - \mathbb{E}[\nabla \ell_{k,\theta^*}^f] \end{bmatrix}\right)$$

$$\xrightarrow{d} \mathcal{N}\left(0, \lambda_k^2 \frac{n}{N_k} \mathrm{Cov}(\nabla \ell_{k,\theta^*}^f)\right) + \mathcal{N}\left(0, \frac{n}{n_k} \mathrm{Cov}(\nabla \ell_{k,\theta^*} - \lambda_k \nabla \ell_{k,\theta^*}^f)\right) \tag{24}$$

$$= \mathcal{N}\left(0, \frac{r}{\tilde{\rho}_k} \cdot V_{k,f,\theta^*}^{\lambda_k} + \frac{1}{\rho_k} V_{k,\Delta,\theta^*}^{\lambda_k}\right). \tag{25}$$

Since $\theta^*$ satisfies $\mathbb{E}[\nabla \ell_{\theta^*}] = 0$, by the law of total expectation

$$\sum_{k=1}^K w_k \mathbb{E}[\nabla \ell_{k,\theta^*}] = \mathbb{E}[\nabla \ell_{\theta^*}] = 0. \tag{26}$$

Using this fact, we can write

$$\nabla L_{\hat{\lambda}}^{\mathrm{SPP}}(\theta) = \sum_{k=1}^K w_k \left(\nabla\left(\hat{\lambda}_k \tilde{L}_{N_k}^f(\theta) + L_{n_k}(\theta) - \hat{\lambda}_k L_{n_k}^f(\theta)\right) - \mathbb{E}[\nabla \ell_{k,\theta^*}]\right). \tag{27}$$

Since samples across the $K$ (where here $K$ is a constant) stratas are independent, combining the results of (25) with (27) yields

$$\sqrt{n}\nabla L_{\hat{\lambda}}^{\mathrm{SPP}}(\theta^*) \xrightarrow{d} \mathcal{N}(0, B_w^\lambda). \tag{28}$$

Applying the mean value theorem, we have

$$\nabla L_{\hat{\lambda}}^{\mathrm{SPP}}(\hat{\theta}_{\hat{\lambda}}^{\mathrm{SPP}}) = \nabla L_{\hat{\lambda}}^{\mathrm{SPP}}(\theta^*) + \nabla^2 L_{\hat{\lambda}}^{\mathrm{SPP}}(\tilde{\theta})(\hat{\theta}_{\hat{\lambda}}^{\mathrm{SPP}} - \theta^*) \tag{29}$$

for some $\tilde{\theta}$ between $\hat{\theta}_{\hat{\lambda}}^{\mathrm{SPP}}$ and $\theta^*$. Let $A^\dagger$ represent the pseudoinverse of $A$. Then

$$\hat{\theta}_{\hat{\lambda}}^{\mathrm{SPP}} - \theta^* = \nabla^2 L_{\hat{\lambda}}^{\mathrm{SPP}}(\tilde{\theta})^\dagger \left(\nabla L_{\hat{\lambda}}^{\mathrm{SPP}}(\hat{\theta}_{\hat{\lambda}}^{\mathrm{SPP}}) - \nabla L_{\hat{\lambda}}^{\mathrm{SPP}}(\theta^*)\right). \tag{30}$$

As $\hat{\theta}_{\hat{\lambda}}^{\mathrm{SPP}} \xrightarrow{p} \theta^*$ per Lemma A.2, we have $\tilde{\theta} \xrightarrow{p} \theta^*$. Under the regularity conditions of Definition A.1 and the fact that $\hat{\lambda}_k \xrightarrow{p} \lambda_k$, an application of the uniform weak law of large numbers and the continuous mapping theorem then gives

$$\nabla^2 L_{\hat{\lambda}}^{\mathrm{SPP}}(\tilde{\theta})^\dagger \xrightarrow{p} \nabla^2 \mathbb{E}[\ell_{\theta*}]^{-1}, \tag{31}$$

which is a constant term. Furthermore, by the law of total expectation,

$$\mathbb{E}[\nabla^2 \ell_{\theta*}]^{-1} = \left(\sum_{k=1}^K w_k H_{k,\theta^*}\right)^{-1} = A_w^{-1}. \tag{32}$$

Finally, the fact that $\hat{\theta}_{\hat{\lambda}}^{\mathrm{SPP}} \xrightarrow{p} \theta^*$ and $\theta^* \in \mathrm{int}(\Theta)$, together with the fact that $\hat{\theta}_{\hat{\lambda}}^{\mathrm{SPP}}$ is a minimum of $L_{\hat{\lambda}}^{\mathrm{SPP}}$, implies that $\nabla L_{\hat{\lambda}}^{\mathrm{SPP}}(\hat{\theta}_{\hat{\lambda}}^{\mathrm{SPP}}) \xrightarrow{p} 0$. Combining these results with (28) via Lemma A.1 and the fact that $A_w^{-1}$ is symmetric for twice continuously differentiable $\ell_\theta$ gives

$$\sqrt{n}(\hat{\theta} - \theta^*) \xrightarrow{d} \mathcal{N}(0, A_w^{-1} B_w^\lambda (A_w^{-1})^\top) = \mathcal{N}(0, A_w^{-1} B_w^\lambda A_w^{-1}). \tag{33}$$

$\square$

## A.2 Proof of Corollary 2

*Proof.* The regularity conditions of Definition A.1 allow us to apply Lemma 4.3 of [20] to show that each of the plug-in estimates for each stratum term satisfies

$$\widehat{H}_{k,\hat{\theta}^{\text{SPP}}} \xrightarrow{p} H_{k,\theta^*} \quad \text{and} \quad \widehat{V}^{\lambda_k}_{k,f,\hat{\theta}^{\text{SPP}}} \xrightarrow{p} \widehat{V}^{\lambda_k}_{k,f,\theta^*} \quad \text{and} \quad \widehat{V}^{\lambda_k}_{k,\Delta,\hat{\theta}^{\text{SPP}}} \xrightarrow{p} \widehat{V}^{\lambda_k}_{k,\Delta,\theta^*}$$

which implies that the weighted plug-in estimate for $\hat{\Sigma}^{\hat{\lambda}}_w$ satisfies $\hat{\Sigma}^{\hat{\lambda}}_w \xrightarrow{p} \Sigma^{\lambda}_w$. (14) is thus equivalent to taking the $\left(\frac{\alpha}{2}, 1 - \frac{\alpha}{2}\right)$ quantiles of the asymptotic normal distribution of $\hat{\theta}^{\text{SPP}}$, implying (15). □

## A.3 Proof of Proposition 1

*Proof.* In the special case of the square loss, the Hessian is the identity matrix. Therefore, simplifying and applying the linearity of the trace, we have

$$\text{Tr}(\Sigma^{\lambda}_w) = \sum_{k=1}^{K} w_k^2 \text{Tr}\left(\frac{r}{\tilde{\rho}_k} \cdot V^{\lambda_k}_{k,f,\theta^*} + \frac{1}{\rho_k} V^{\lambda_k}_{k,\Delta,\theta^*}\right) \tag{34}$$

$$= \sum_{k=1}^{K} w_k \text{Tr}\left(r \cdot V^{\lambda_k}_{k,f,\theta^*} + V^{\lambda_k}_{k,\Delta,\theta^*}\right) \tag{35}$$

$$= r \sum_{k=1}^{K} w_k \text{Tr}\left(V^{\lambda_k}_{k,f,\theta^*}\right) + \sum_{k=1}^{K} w_k \text{Tr}\left(V^{\lambda_k}_{k,\Delta,\theta^*}\right) \tag{36}$$

$$= r \sum_{j=1}^{d} \sum_{k=1}^{K} w_k V^{\lambda_k}_{k,f,\theta^*,jj} + \sum_{j=1}^{d} \sum_{k=1}^{K} w_k V^{\lambda_k}_{k,\Delta,\theta^*,jj} \tag{37}$$

where we include the subscript $jj$ to index the diagonal variance terms of both covariance matrices. For ease of notation, denote the conditional variances as

$$V_{f,j} \mid Z = k := \lambda^2 \nabla \ell_{\theta_j^*}(X, f(X)) \mid X \in \mathcal{A}_k \quad \text{and}$$
$$V_{\Delta,j} \mid Z = k := \nabla \ell_{\theta_j^*}(X,Y) - \lambda \nabla \ell_{\theta_j^*}(X, f(X)) \mid X \in \mathcal{A}_k.$$

Then we can write (37) as

$$r \sum_{j=1}^{d} \sum_{k=1}^{K} w_k V^{\lambda_k}_{k,f,\theta^*,jj} + \sum_{j=1}^{d} \sum_{k=1}^{K} w_k V^{\lambda_k}_{k,\Delta,\theta^*,jj} = \sum_{j=1}^{d} r\mathbb{E}[\text{Var}(V_{f,j} \mid Z)] + \mathbb{E}[\text{Var}(V_{\Delta,j} \mid Z)], \tag{38}$$

and apply the law of total variance to get

$$\text{Tr}(\Sigma^{\lambda}_w) = \sum_{j=1}^{d} r\mathbb{E}[\text{Var}(V_{f,j} \mid Z)] + \mathbb{E}[\text{Var}(V_{\Delta,j} \mid Z)] \tag{39}$$

$$= \sum_{j=1}^{d} r(\text{Var}(V_{f,j}) - \text{Var}(\mathbb{E}[V_{f,j} \mid Z])) + (\text{Var}(V_{\Delta,j}) - \text{Var}(\mathbb{E}[V_{\Delta,j} \mid Z])) \tag{40}$$

$$\leq \sum_{j=1}^{d} r\text{Var}(V_{f,j}) + \text{Var}(V_{\Delta,j}) \tag{41}$$

$$= \text{Tr}(\Sigma^{\lambda}). \tag{42}$$

Finally, (41) holds with equality iff both $\text{Var}(\mathbb{E}[V_{f,j} \mid Z]) = 0$ and $\text{Var}(\mathbb{E}[V_{\Delta,j} \mid Z]) = 0$, which, combined with the assumption that $\mathbb{P}(Z = k) = w_k > 0$ for all $k$, is only satisfied when both $\mathbb{E}[V_{f,j} \mid Z]$ and $\mathbb{E}[V_{\Delta,j} \mid Z]$ are constants. □

### A.4 Proof of Proposition 2

*Proof.* By linearity of $B_w^\lambda$, we can rewrite $\Sigma_w^\lambda = A_w^{-1} B_w^\lambda A_w^{-1}$ as

$$\Sigma_w^\lambda = \sum_{k=1}^K w_k^2 A_w^{-1} \left( \frac{r}{\tilde{\rho}_k} \cdot V_{k,f,\theta^*}^{\lambda_k} + \frac{1}{\rho_k} V_{k,\Delta,\theta^*}^{\lambda_k} \right) A_w^{-1} \tag{43}$$

$$= \sum_{k=1}^K \frac{w_k^2}{\rho_k} A_w^{-1} \left( \frac{r\rho_k}{\tilde{\rho}_k} \cdot V_{k,f,\theta^*}^{\lambda_k} + V_{k,\Delta,\theta^*}^{\lambda_k} \right) A_w^{-1} \tag{44}$$

$$= \sum_{k=1}^K \frac{w_k^2}{\rho_k} A_w^{-1} \left( \frac{n_k}{N_k} \cdot V_{k,f,\theta^*}^{\lambda_k} + V_{k,\Delta,\theta^*}^{\lambda_k} \right) A_w^{-1}. \tag{45}$$

By linearity of the trace, we then also have

$$\mathrm{Tr}(\Sigma_w^\lambda) = \sum_{k=1}^K \frac{w_k^2}{\rho_k} \mathrm{Tr} \left( A_w^{-1} \left( \frac{n_k}{N_k} \cdot V_{k,f,\theta^*}^{\lambda_k} + V_{k,\Delta,\theta^*}^{\lambda_k} \right) A_w^{-1} \right). \tag{46}$$

As each term in the sum is independent, we can minimize the sum by minimizing each individual term, i.e.,

$$\lambda_k^* = \operatorname*{argmin}_{\lambda_k} \mathrm{Tr} \left( A_w^{-1} \left( \frac{n_k}{N_k} \cdot V_{k,f,\theta^*}^{\lambda_k} + V_{k,\Delta,\theta^*}^{\lambda_k} \right) A_w^{-1} \right) \tag{47}$$

for $k = 1, \ldots, K$. We can then directly apply Proposition 2 of [3] to get

$$\lambda_k^* = \frac{\mathrm{Tr} \left( A_w^{-1} (\mathrm{Cov}(\nabla \ell_{k,\theta^*}, \nabla \ell_{k,\theta^*}^f) + \mathrm{Cov}(\nabla \ell_{k,\theta^*}^f, \nabla \ell_{k,\theta^*})) A_w^{-1} \right)}{2(1 + r_k) \mathrm{Tr} \left( A_w^{-1} \mathrm{Cov}(\nabla \ell_{k,\theta^*}^f) A_w^{-1} \right)}, \tag{48}$$

where $r_k = n_k / N_k$. $\qquad\square$

### A.5 Proof of Proposition 3

*Proof.* Following the same derivation as Proposition 2, we can rewrite

$$\mathrm{Tr}(\Sigma_w^\lambda) = \sum_{k=1}^K w_k^2 \mathrm{Tr} \left( A_w^{-1} \left( \frac{r}{\tilde{\rho}_k} \cdot V_{k,f,\theta^*}^{\lambda_k} + \frac{1}{\rho_k} V_{k,\Delta,\theta^*}^{\lambda_k} \right) A_w^{-1} \right) \tag{49}$$

$$= r \sum_{k=1}^K \frac{w_k^2}{\tilde{\rho}_k} \mathrm{Tr} \left( A_w^{-1} V_{k,f,\theta^*}^{\lambda_k} A_w^{-1} \right) + \sum_{k=1}^K \frac{w_k^2}{\rho_k} \mathrm{Tr} \left( A_w^{-1} V_{k,\Delta,\theta^*}^{\lambda_k} A_w^{-1} \right), \tag{50}$$

and therefore can optimize $\tilde{\rho}_k$ and $\rho_k$ independently.

We start with $\rho_k$. Let $z_k = \mathrm{Tr} \left( A_w^{-1} V_{k,\Delta,\theta^*}^{\lambda_k} A_w^{-1} \right)$. We then solve the constrained optimization problem

$$\text{minimize} \ \sum_{k=1}^K \frac{w_k^2}{\rho_k} z_k \quad \text{s.t.} \quad \sum_{k=1}^K \rho_k = 1, \rho_k \geq 0. \tag{51}$$

Turning this into a Lagrangian with additional slack variables $s_k \geq 0$, we have

$$\mathcal{J}(\rho, \mu, \eta, s) = \sum_{k=1}^K \frac{w_k^2}{\rho_k} z_k + \mu \left( \sum_{i=1}^K \rho_k - 1 \right) + \sum_{k=1}^K \eta_k (\rho_k - s_k^2) \tag{52}$$

Setting $\nabla \mathcal{J}(\rho, \mu, \eta) = 0$ and solving for $\rho_k$ then gives:

$$\frac{\partial \mathcal{J}}{\partial \rho_k} = -\frac{w_k^2}{\rho_k^2} z_k + \mu + \eta_k = 0 \tag{53}$$

$$\frac{\partial \mathcal{J}}{\partial \mu} = 1 - \sum \rho_k = 0 \tag{54}$$

$$\frac{\partial \mathcal{J}}{\partial \eta_k} = \rho_k - s_k^2 = 0 \tag{55}$$

$$\frac{\partial \mathcal{J}}{\partial s_k} = -2\eta_k s_k = 0 \tag{56}$$

Assume that the inequality constraint is inactive, and $\eta_k = 0$. Solving for $\rho_k^*$,

$$\rho_k^* = s_k^2 = \sqrt{\frac{w_k^2 z_k}{\mu}} \implies \sum_{k=1}^{K} \sqrt{\frac{w_k^2 z_k}{\mu}} = 1 \tag{57}$$

$$\implies \sqrt{\mu} = \sum_{k=1}^{K} \sqrt{w_k^2 z_k} \tag{58}$$

$$\implies \rho_k^* = \frac{w_k \sqrt{z_k}}{\sum_{k'=1}^{K} w_{k'} \sqrt{z_{k'}}}. \tag{59}$$

We can now verify that the constraint is inactive, with $\rho_k^* \geq 0$, since $\sum_{k=1}^{K} w_k = 1, w_k \geq 0$ by definition of a valid probability distribution, and we have $z_k \geq 0$ since it is a sum of non-negative variance terms.

The same analysis can then be applied to $\tilde{\rho}_k$, but for $z_k = \mathrm{Tr}\left(A_w^{-1} V_{k,f,\theta^*}^{\lambda_k} A_w^{-1}\right)$. $\qquad\square$

## B    A simplified estimate of the sample allocation for mean estimation

In the setting of Example 2 (i.e., 1-$d$ mean estimation), assume that $c(y \mid x) \approx \mathbb{P}(Y = y \mid X = x)$ is a confidence estimate for label $y \in \mathcal{Y}$, where $\mathcal{Y}$ is discrete. Then, if we define the autorater as

$$f(x) = \sum_{y \in \mathcal{Y}} c(y \mid x) \cdot y \approx \mathbb{E}[Y \mid X = x], \tag{60}$$

the term $\mathrm{Var}(Y - \lambda_k f(X) \mid X \in \mathcal{A}_k)$ further simplifies to $\approx \mathrm{Var}(Y \mid X \in \mathcal{A}_k)$ for any value of $\lambda_k$. Applying the law of total variance, we can conveniently write $\mathrm{Var}(Y \mid X \in \mathcal{A}_k)$ as

$$\mathrm{Var}(Y \mid X \in \mathcal{A}_k) = \mathbb{E}[\mathrm{Var}(Y \mid X) \mid X \in \mathcal{A}_k] + \mathrm{Var}(\mathbb{E}[Y \mid X] \mid X \in \mathcal{A}_k), \tag{61}$$

which can be empirically estimated as $\hat{\sigma}_k$ on unlabeled autorater data, $\tilde{X}_{ik}, i = 1, \dots, N_k$, via

$$\hat{\sigma}_k^2 = \widehat{\mathbb{E}}_{N_k}\left[\sum_{y \in \mathcal{Y}} c(y \mid \tilde{X}_{ik}) \cdot y - f(\tilde{X}_{ik})\right] + \widehat{\mathrm{Var}}_{N_k}(f(\tilde{X}_{ik})), \tag{62}$$

where $\widehat{\mathbb{E}}_{N_k}$ and $\widehat{\mathrm{Var}}_{N_k}$ denote the empirical mean and variance over all $\tilde{X}_{ik}$, respectively. Lastly, when $\mathcal{Y}$ is binary (which is the case in all of our experiments in §5.2), this becomes

$$\hat{\sigma}_k^2 = \widehat{\mathbb{E}}_{N_k}\left[f(\tilde{X}_{ik})(1 - f(\tilde{X}_{ik}))\right] + \widehat{\mathrm{Var}}_{N_k}(f(\tilde{X}_{ik})), \tag{63}$$

which can be easily calculated in Python as `np.mean(yhat * (1 - yhat)) + np.var(yhat)`.

## C    Additional experimental results

We also evaluate `StratPPI` on the Chatbot Arena dataset [10], in which we evaluate the win-rate of `gpt-4-1106-preview` over `claude-2.1`.[4] See Figure 3. Specifically, we use the standard Chatbot

---

[4]https://huggingface.co/datasets/lmsys/lmsys-arena-human-preference-55k

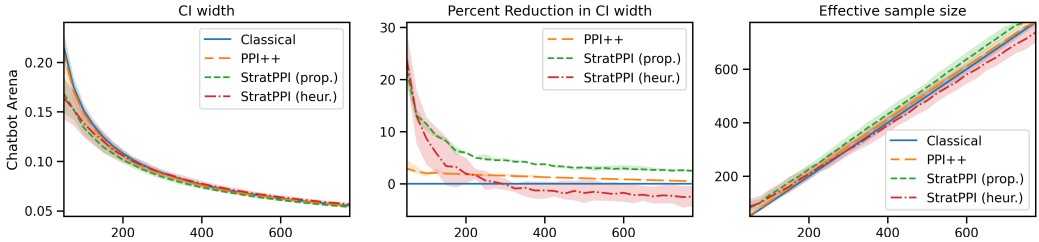

Figure 3: Win-rate experiment on Chatbot Arena for `gpt-4-1106-preview` vs. `claude-2.1`. Scores are based on the average label ('better' = 1 vs. 'worse' = 0) over 10 samples from Gemini Ultra acting as a LLM judge. Interestingly, as these confidence scores are not calibrated, our heuristic becomes overly aggressive at higher $n$. Future work can explore how to best incorporate additional regularization into the estimated optimal sampling ratios $\rho$.

Arena auto-eval prompt[5] and map `[[A≫B]]`, `[[A>B]]` to 1, `[[A=B]]` to 0.5, and `[[B≫A]]`, `[[B>A]]` to 0. We then do self-consistency sampling and take the average of 10 samples from Gemini Ultra. This is used as both our final autorater estimate and confidence. We find that stratification also helps in this setting. However, in line with prior and contemporary work [e.g., 35; 16] we found confidence scores from the LLM-as-a-judge to be fairly uncalibrated, which makes our heuristic allocation less effective at larger n (though still effective at small labeled sample sizes $n$). Investigating robust heuristics in the face of miscalibration (e.g., via regularization or recalibration) would likely help, and is a direction for future work.

---
[5]https://github.com/lm-sys/arena-hard-auto/blob/main/config/judge_config.yaml

