# OpenReview forum: "Stratified Prediction-Powered Inference for Effective Hybrid Evaluation of Language Models"
_NeurIPS.cc/2024/Conference — NeurIPS 2024 poster_

### Official Review · Reviewer_deoY · 2024-07-12

**Soundness:** 3
**Presentation:** 3
**Contribution:** 3
**Rating:** 6
**Confidence:** 3

**Summary:**

Taking inspiration from stratified sampling, this paper proposes a method named Stratified Prediction-Powered Inference (StratePPI).
With appropriate choices of stratification and sample allocation, StratePPI can provide substantially tighter confidence intervals than unstratified approaches.  The experimental results on simulated and real data show the effectiveness of StratePPI.

**Strengths:**

1. This paper introduces data stratification strategies to improve prediction-powered inference (PPI) for the first time.
2. The experimental results show that StratPPI is more effective than PPI++.

**Weaknesses:**

The stratification is very important to StratPPI, and there is little introduction on how to stratify data effectively in this paper.

**Questions:**

In practical applications, how should we stratify data to ensure that StratPPI is better than PPI or PPI++?

**Limitations:**

The authors have adequately addressed the limitations.

---

> ### Author Rebuttal · Authors · 2024-08-07
>
> We thank the reviewer for their careful review and thoughtful comments. We provide answers to specific questions and remarks (quoted) below.
>
> > In practical applications, how should we stratify data to ensure that StratPPI is better than PPI or PPI++?
>
> In general we should stratify data so that examples with similar expected labels are grouped together (e.g., the data should be relatively homogeneous within each stratum). This reduces the variance of the stratified estimate. This can then be expected to help when the data pre-stratification is not homogenous. In the experiments presented here, we found that equal mass partitioning based on confidence scores reliably satisfied these properties (and, at least at the extreme confidence levels it makes sense to expect that labels will have low variance for any reasonably well-calibrated classifier).
>
> Please let us know if this has addressed your concerns. We look forward to engaging in the response period.

---

> > ### Comment · Reviewer_deoY · 2024-08-12
> >
> > Glad to hear your explanation; it is indeed a commendable research effort.

---

### Official Review · Reviewer_GJBL · 2024-07-14

**Soundness:** 3
**Presentation:** 3
**Contribution:** 3
**Rating:** 6
**Confidence:** 3

**Summary:**

The paper proposes an extension of prediction-powered inference (PPI) - a method for calculating confidence intervals with narrower confidence bands. The data is separated into subsets and the bias rectification term in the confidence calculation is weighted based on the similarity to the particular subsets. Evaluation is performed on one synthetic dataset and 4 real datasets, showing narrower confidence bands if the data is heterogeneous.

**Strengths:**

The paper is well written. Even the necessary background, regarding existing work on PPI, is explained well enough to understand.
The adjustment of the calculation based on data similarity makes sense.
Empirical evaluation on several datasets is good to see.

**Weaknesses:**

Line 107 defines Yi as the target output, which in the case of QA would normally be a textual answer. Line 112 then defines Y as the binary gold rating of correctness of the answer. This is contradictory and causes confusion when following the other equations.

The chosen evaluation datasets don't seem to be particularly widely used. Evaluation on some more common QA datasets would have been more impactful.

It is unclear whether this method actually helps in calculating more accurate means, or whether it only produces narrower confidence bands. And if it is the latter, then can you give more details on the practical benefits? It's a different method of calculating the confidence bands so the bands would not be directly comparable to other methods and couldn't really say that the new ones are "better" than the previous ones.

**Questions:**

Please see above

**Limitations:**

Yes

---

> ### Author Rebuttal · Authors · 2024-08-07
>
> We thank the reviewer for their careful review and thoughtful comments. We provide answers to specific questions and remarks (quoted) below.
>
>
> > Line 107 defines $Y_i$ as the target output, which in the case of QA would normally be a textual answer. Line 112 then defines $Y$ as the binary gold rating of correctness of the answer.
>
> Thanks for pointing out this confusion. The task here is _evaluating_ the correctness of a QA system, and not inferring the answer itself. As such, $Y_i \in {0, 1}$ is the target output of the _evaluation_ task (i.e., incorrect vs. correct). The input $X_i$ to the autorater for this evaluation task is the pair (QA question, model answer). We will make this clearer in the text.
>
>
> > The chosen evaluation datasets don't seem to be particularly widely used.
>
> We chose to evaluate on high-quality, well-cited datasets that also have pre-existing autorater scores. In the supplement to this rebuttal, we have also included results on the Chatbot Arena dataset.
>
> > It is unclear whether this method actually helps in calculating more accurate means, or whether it only produces narrower confidence bands.
>
> Our stratified PPI estimate reduces variance. This reduces the size of the confidence band, which is the main focus of the paper. Narrower confidence bands allow practitioners to get more precise error estimates with fewer annotated samples. This is very important to do given the high costs (in terms of both time and money) of running human evaluations. Additionally, since the PPI estimate is unbiased, this also has the effect of reducing the mean squared error of the estimate, which can be seen from the usual bias-variance decomposition of the error. To illustrate this, we have also plotted the RMSE of our estimator on the task in Figure 1 of the paper in the rebuttal supplement (see Figure S.1). The trends of the confidence interval width and the RMSE are identical.
>
> Please let us know if this has addressed your concerns. We look forward to engaging in the response period.

---

> > ### Comment · Reviewer_GJBL · 2024-08-09
> > **Thanks**
> >
> > Thanks for clarifying. You can't really use the same symbol for very different things 10 lines apart without redefining it first, so I would suggest changing that.
> >
> > Additional results are good, although you could have included a citation or source to show exactly which version of the dataset you are using.
> >
> > I am still unsure about the practical benefits. The provided justification sounds analogous to getting non-significant results using a statistical significance calculation, then switching to a different significance calculation method that indicates a significant result. Even though the measurement changes, the actual data and the real result stay the same. So the benefit should be somewhere else.
> >
> > This is somewhat outside of my field of expertise, which I have also indicated with the low confidence score, and I'm not comfortable raising my score higher than it currently is.

---

> > > ### Author Response · Authors · 2024-08-09
> > > **Clarifying a misunderstanding**
> > >
> > > We thank the reviewer for their reply to our rebuttal, and appreciate the opportunity to engage.
> > >
> > >
> > > > You can't really use the same symbol for very different things 10 lines apart without redefining it first, so I would suggest changing that.
> > >
> > >
> > > To restate our response from earlier: the definition of Y_i does not change. Y_i is always the target output for the autorater. The target output for the autorater is always a measure of quality for the input X_i (e.g., if it is correct, helpful, toxic, etc). Y_i never refers to the  target output of the LLM-based QA system, this is always part of X (see  L111). The autorater represents a separate prediction problem from the QA system it is evaluating. We do appreciate the reviewer’s confusion, and will definitely make the problem statement clearer and the distinction explicit.
> > >
> > >
> > > > Additional results are good, although you could have included a citation or source to show exactly which version of the dataset you are using.
> > >
> > >
> > > Our apologies for the oversight in the rebuttal! Thanks for pointing this out. We used the dataset available at https://huggingface.co/datasets/lmsys/lmsys-arena-human-preference-55k. The citation for which is:
> > > ```
> > > @misc{chiang2024chatbot,
> > >     title={Chatbot Arena: An Open Platform for Evaluating LLMs by Human Preference},
> > >     author={Wei-Lin Chiang and Lianmin Zheng and Ying Sheng and Anastasios Nikolas Angelopoulos and Tianle Li and Dacheng Li and Hao Zhang and Banghua Zhu and Michael Jordan and Joseph E. Gonzalez and Ion Stoica},
> > >     year={2024},
> > >     eprint={2403.04132},
> > >     archivePrefix={arXiv},
> > >     primaryClass={cs.AI}
> > > }
> > > ```
> > >
> > >
> > > Of course, this will be properly specified and cited in our final revision.
> > >
> > >
> > > > I am still unsure about the practical benefits. The provided justification sounds analogous to getting non-significant results using a statistical significance calculation, then switching to a different significance calculation method that indicates a significant result. Even though the measurement changes, the actual data and the real result stay the same. So the benefit should be somewhere else.
> > >
> > >
> > > There is a substantial misunderstanding here.
> > >
> > >
> > > "the measurement changes, the actual data... stay the same"
> > >
> > >
> > > This is false. The point of stratified PPI is that the data collection strategy has changed. There is no repeated testing of any kind: it is simply an estimator computed on a strategically partitioned and collected dataset. You can think of Stratified PPI as analogous to experimental design in statistics: deciding which parts of X-space to oversample/undersample in order to get maximum-precision estimates. To make it possible to aggregate the samples taken from each strata, Stratified PPI also benefits from being able to effectively use additional unlabeled data not only for making predictions, but also for estimating the propensity weights for the partitions of this X-space.
> > >
> > >
> > > To keep it simple, the reason for improvements is simple: the estimator computed on this data is __lower-variance__ than the standard PPI estimator. The resulting practical benefit is then also simple: we can get better point estimates (lower MSE) with higher confidence (smaller confidence intervals). This allows for better evaluations, and decision-making based on those evaluations, at less cost.

---

### Official Review · Reviewer_BR9i · 2024-07-15

**Soundness:** 4
**Presentation:** 4
**Contribution:** 3
**Rating:** 7
**Confidence:** 1

**Summary:**

This paper introduces the stratified PPI method, designed to reduce evaluation bias from autoraters by aligning them with a few human annotations. The key novelty lies in identifying that the autorater's performance varies across different example groups, with biases being smaller for some examples and larger for others. Consequently, they propose a straightforward method to group examples into strata and calculate stratum-specific rectification parameters. Experiments demonstrate that stratified PPI consistently achieves tighter confidence intervals compared to baselines.

**Strengths:**

The paper introduces a novel method, stratified PPI, which effectively addresses the issue of evaluation bias in autoraters by considering the varying performance across different groups of examples. The proposed method of grouping examples into strata and calculating stratum-specific rectification parameters is both simple and effective.

**Weaknesses:**

The evaluations are conducted on simple evaluation problems where the evaluation is binary, while current LLM autoraters use a more fine-grained scales (e.g., 1-5). Additionally, real human evaluations may be multinomial, where people's opinion will diverge on the same instance. I'm not sure how this proposed method will perform in that scenario.

**Questions:**

See weaknesses.

**Limitations:**

The authors have adequately addressed the limitations.

---

> ### Author Rebuttal · Authors · 2024-08-07
>
> We thank the reviewer for their careful review and thoughtful comments. We provide answers to specific questions and remarks (quoted) below.
>
> > The evaluations are conducted on simple evaluation problems where the evaluation is binary.
>
> Our Stratified PPI method works for general M-estimation problems, which also includes non-binary mean estimates. Our supplement to this rebuttal also includes results for estimating Bradley-Terry coefficients, another M-estimation problem. The autoraters can also output scores (e.g., 1 to 5) that are different from the human labels (see line 110). In fact, the autoraters we used in our experiments output continuous scores in (0, 1).
>
> > Additionally, real human evaluations may be multinomial, where people's opinion will diverge on the same instance.
>
> Once again, while we focus on simple mean estimation in our experiments, our Stratified PPI method applies to a broad class of general M-estimation problems. This includes multiclass logistic regression, which could serve as a good model for the proposed setting. For details, see [1].
>
> Another interesting situation (not covered here—but important for future work!) are problems in which the human annotators are not just variable, but actually systematically biased with respect to the ground truth (e.g., when the annotators are crowd-workers vs. experts or the target users).
>
> Please let us know if this has addressed your concerns. We look forward to engaging in the response period.
>
> [1] Angelopoulos et. al. PPI++: Efficient Prediction-Powered Inference. 2023.

---

### Official Review · Reviewer_5dNx · 2024-07-16

**Soundness:** 3
**Presentation:** 4
**Contribution:** 4
**Rating:** 7
**Confidence:** 3

**Summary:**

The paper extends the predictive power inference methods to leverage data stratification strategies, and demonstrates that stratification can be used to obtain unbiased statistical estimates, while reducing variance (with worst case being as bad as prior approaches PPI++, and in practice considerably better). Concretely, the authors

1. Demonstrate that for the stratified prediction-powered loss, the minimizer has an asymptotically normal distribution with the true mean, thereby allowing for construction of confidence intervals leveraging stratum conditioned covariance matrices.
2. Provide an method for choosing the optimal weighting between the human annotations and autograder annotations for the stratified setting.
3. Provide the closed form solution for optimal allocation for stratum weights for the labeled and unlabeled data. Furthermore, for the special case of mean estimation, the authors also provide a heuristic way of estimating these weights based on the autograders' confidence for the scores.
4. Demonstrate for simulations that for cases where the strata are homogeneous, the estimators are at worse as bad as PPI++, while in cases where the strata are heterogeneous, StratPPI outperforms PPI++.
5. Finally, the authors demonstrate the utility of the proposed approach on a real-world multilingual summarization benchmark, an attributed QA dataset as well as the galaxy dataset, demonstrating that StratPPI obtains tighter confidence intervals compared to other approaches.

**Strengths:**

1. The proposed method demonstrably produces tighter confidence interval, leveraging data stratification. Such stratification is quite natural to occur in generally available data, in my opinion. So the proposed extension should be very widely applicable in a real world setting.
2. The paper is relatively easy to follow even for someone who is not very familiar with this subdomain, and is quite self contained.

**Weaknesses:**

Given the focus of the paper on effective evaluation of language models, it would have been good for the paper to have a stronger focus on evaluation of the numerous publicly available (large) language models. Concretely, imo, the proposed method should be extendible to assessing pairwise relative performance of language models (similar to Section 3.1 of [1]). It would be good to compare confidence intervals for the BT coefficients obtained from PPI++ vs StratPPI for (a subset of) different Language Models on the Chatbot arena.

[1] Boyeau, Pierre, et al. "AutoEval Done Right: Using Synthetic Data for Model Evaluation." arXiv preprint arXiv:2403.07008 (2024).

**Questions:**

For Figure 1, column 3, what is the intuition about why is the coverage of StratPPI (prop) so low compared to all other counterparts when n is small ?

---

> ### Author Rebuttal · Authors · 2024-08-07
>
> We thank the reviewer for their careful review and thoughtful comments. We provide answers to specific questions and remarks (quoted) below.
>
> > It would have been good for the paper to have a stronger focus on evaluation of the numerous publicly available (large) language models.
>
> As suggested, we have included an additional experiment in Figure S.1 of the supplement to this rebuttal for estimating win-rates between gpt-4-1106-preview and claude-2.1 on the Chatbot Arena dataset. Specifically, we use the standard Chatbot Arena auto-eval prompt (see https://github.com/lm-sys/arena-hard-auto/blob/main/config/judge_config.yaml), and map `[[A>>B]]`, `[[A>B]` to 1, `[[A=B]]` to 0.5, and `[[B>>A]]`, `[[B>A]]` to 0. We then do self-consistency sampling and take the average of 10 samples from Gemini Ultra. This is used as both our final autorater estimate and confidence. We find that stratification also helps in this setting. As an interesting side note, in line with previous work we found confidence scores from the LLM-as-a-judge to be fairly uncalibrated, which makes our heuristic allocation less effective at larger n (though still effective at small n). Investigating more robust heuristics in the face of miscalibration (e.g., via regularization) would likely help. Proportional allocation is effective at all values of n. We will also include results on the larger multiple model setting when using Bradley-Terry model coefficients for the final version of the paper.
>
>
> > What is the intuition about why is the coverage of StratPPI (prop) so low compared to all other counterparts when n is small?
>
> The difference in coverage reported is small, and is an artifact of a smaller number of trials run for the coverage estimation. When increasing the number of simulations for estimating coverage to 5000, we observe a more stable coverage result. This is included in Figure S.1 of the rebuttal supplement. (Note that we also fixed column 1 from the submitted manuscript, which had been run at the wrong settings.)
>
> Please let us know if this has addressed your concerns. We look forward to engaging in the response period.

---

> > ### Comment · Reviewer_5dNx · 2024-08-09
> > **Response to rebuttal**
> >
> > The additional experiments address my concerns. If possible, it would be good for the results for the LM evaluations to be included in the main paper instead of being relegated to the appendix.
> > I have increased the confidence for my review. In my opinion, this work is pretty solid.

---

> > > ### Author Response · Authors · 2024-08-09
> > >
> > > Thank you for the review and the response to our rebuttal! We agree that the LM evaluation results should be included in the main paper, and will make sure to put them there.

---

### Author Rebuttal · Authors · 2024-08-07

Thank you to all the reviewers for taking the time to read and comment on our work. We were pleased to receive several good suggestions, and have taken this feedback into account. Please see our uploaded pdf for supplemental results.

Specifically, we have added another very realistic experiment on the Chatbot Arena dataset, in which we measure the win-rate between `gpt-4-1106-preview` and `claude-2.1`. Details are included in the figure caption. We have also included additional results on our illustrative synthetic task that shows how the MSE of the prediction powered point estimate and the size of the confidence intervals follow similar trends. Stratified PPI improves on both CI size and MSE.

We have responded to the remainder of the comments raised by reviewers individually. Please let us know if any questions or concerns remain. We look forward to additional discussion.

---

### Decision · Program_Chairs · 2024-09-25

**Decision:**

Accept (poster)

**Comment:**

This paper presents the stratified PPI method, aimed at minimizing evaluation bias from autoraters by calibrating them against a limited number of human annotations. One particular innovation is the recognition that an autorater's accuracy differs across various groups of examples, with some groups exhibiting smaller biases and others showing larger ones. Based on this insight, the authors propose a simple approach to categorize examples into distinct strata and compute adjustment parameters for each stratum. Experimental results indicate that the stratified PPI method consistently produces more precise confidence intervals than existing baseline methods.

While the reviewers see the merit in the paper, a repeated weakness has to do with the evaluation: the number of LLMs evaluated the nature of the evaluation task and data etc. While this is something to be addressed, and the authors recognize that and have even added an experiment, it seems that the reviewers are willing to accept the paper in its current form and I respect that.